# Modeling the relative risk of SARS-CoV-2 infection to inform risk-cost-benefit analyses of activities during the SARS-CoV-2 pandemic

**John E. McCarthy**[1]*, **Barry D. Dewitt**[2], **Bob A. Dumas**[3], **Myles T. McCarthy**[4]

**1** Department of Mathematics and Statistics, Washington University in St. Louis, Saint Louis, Missouri, United States of America, **2** Department of Engineering & Public Policy, Carnegie Mellon University, Pittsburgh, PA, United States of America, **3** Omnium LLC, Saint Joseph, MO, United States of America, **4** University of Illinois at Urbana-Champaign, Champaign, IL, United States of America

\* mccarthy@wustl.edu

**Data Availability Statement:** All relevant data are within the manuscript and its Supporting information files.

## Abstract

Risk-cost-benefit analysis requires the enumeration of decision alternatives, their associated outcomes, and the quantification of uncertainty. Public and private decision-making surrounding the COVID-19 pandemic must contend with uncertainty about the probability of infection during activities involving groups of people, in order to decide whether that activity is worth undertaking. We propose a model of SARS-CoV-2 infection probability that can produce estimates of relative risk of infection for diverse activities, so long as those activities meet a list of assumptions, including that they do not last longer than one day (e.g., sporting events, flights, concerts), and that the probability of infection among possible routes of infection (i.e., droplet, aerosol, fomite, and direct contact) are independent. We show how the model can be used to inform decisions facing governments and industry, such as opening stadiums or flying on airplanes; in particular, it allows for estimating the ranking of the constituent components of activities (e.g., going through a turnstile, sitting in one's seat) by their relative risk of infection, even when the probability of infection is unknown or uncertain. We prove that the model is a good approximation of a more refined model in which we assume infections come from a series of independent risks. A linearity assumption governing several potentially modifiable risks factors—such as duration of the activity, density of participants, and infectiousness of the attendees—makes interpreting and using the model straightforward, and we argue that it does so without significantly diminishing the reliability of the model.

## 1 Introduction

Coronavirus disease 2019 (COVID-19), caused by severe acute respiratory syndrome-coronavirus 2 (SARS-CoV-2), has caused a pandemic. As of November 6, 2020, the Johns Hopkins University COVID-19 dashboard reports approximately 49 million cases and 1.2 million deaths due to the disease [1, 2]. Social distancing and shutting businesses have reduced the number of cases, but there is mounting pressure to reopen businesses. The purpose of this paper is to provide a model to estimate the relative infection risks of different activities. That

**Funding:** JEM was partially supported by National Institutes of Health Grant R01 AG052550-01A1 and National Science Foundation Grant DMS 1565243. BDD was supported by the Riksbanken Jubileumsfond program on Science and Proven Experience (M14-0138:1). JEM, BAD, and MTM received funding through Omnium LLC in Omnium's capacity as a consultant for Delaware North, a company that may be affected by the research reported in the paper. The funders provided support in the form of salaries for authors, but did not have any additional role in the study design, data collection and analysis, decision to publish, or preparation of the manuscript. The specific roles of these authors are articulated in the 'author contributions' section.

**Competing interests:** BAD is the owner of W. Etienne Dumas Co., an S Corporation in the state of Washington that is a vendor to Omnium LLC. JEM is the owner of E M C Squared and MTM is a vendor to E M C Squared, which is a vendor to Omnium LLC. JEM, BAD, and MTM received funding through Omnium LLC in Omnium's capacity as a consultant for Delaware North, a company that may be affected by the research reported in the paper. There are no patents, products in development or marketed products associated with this research to declare. This does not alter our adherence to PLOS ONE policies on sharing data and materials.

information can allow decision-makers in industry and government to rank activities according to their relative risk of infection. In combination with an understanding of the benefits and costs of those activities, decision-makers can then make informed choices about whether, and if so, how to allow participation in previously forbidden activities.

Despite much ongoing research, there are many parameters of coronavirus disease that remain uncertain, such as the effective reproduction number of the virus given various characteristics of a population, or the precise effectiveness of various non-pharmaceutical interventions, or the significance of aerosol transmission [3–6]. Whereas much effort has been focused on determining these and other characteristics, many of which are needed to produce estimates of absolute risk of infection, such estimates are still uncertain. Nonetheless, policy decisions need to be made.

Risk-cost-benefit analysis provides one framework with which to analyze policy alternatives in order to inform policy decisions. In general terms, it aims to characterize the undesirable outcomes and the probabilities of those outcomes (i.e., the risks) for each decision alternative, the possibly uncertain costs of those alternatives, and their possibly uncertain benefits [7]. In its approach informed by behavioral decision research [7–9], the process involves not just normative analysis but also analysis to understand how the public perceives of the alternatives (i.e., descriptive analysis), and how to bridge the normative and descriptive perspective, when they differ (i.e., prescriptive analysis). Ultimately, the decision-maker also needs to perform a decision analysis with all of the information they have collected, which involves deciding on some decision rule to choose among the alternatives as characterized by their respective risks, costs, benefits, and the associated uncertainties.

In this study, we propose a model to estimate the relative risk of SARS-CoV-2 infection that we believe is useful for characterizing that risk for a large set of activities in both the private sector (e.g., attending a concert) and public sector (e.g., accessing government services in-person). That characterization also illuminates modifiable factors that can lower the risk of infection of a given activity. In combination with other information about the benefits and costs, the model provides a useful tool for anyone undertaking risk-cost-benefit analyses during the pandemic.

More specifically, we propose that when planning for activities that last no more than one day, we can use a model of infection probability that is linear in many potentially controllable variables, such as duration of the activity, density of participants, and infectiousness rate among the attendees. The advantages of that linearity are that it greatly simplifies analyses of different scenarios (for example, the effects of reducing density, or reducing the time spent in specific activities), and also allows comparison of relative risks across different events, even when the base parameters needed to estimate absolute risk are unknown.

This paper is organized as follows: in Section 2, we describe the assumptions of the model, and describe the model mathematically. In Section 3 we present several example calculations, analyzing the risks of idealized versions of airplane travel, attending a sporting event, sitting in a classroom, going to a restaurant, and attending a religious service. In Section 4 discuss how the model is useful, provide guidance for how it might be used, and address its limitations. Appendices provide mathematical details that we exclude from the main text, and an Excel program in the online Supplemental Information provides more detailed calculations for the paper's examples.

## 2 The model

Here, we describe our model. We believe it is important to have a robust and well-formed, mechanistic, model of infection transmission, even if it contains many unknown parameters.

As we will argue, this will allow us to draw inferences on *relative risks* even if we cannot quantify *absolute risks*. The availability of a model of infection transmission that is mechanistic will allow for comparative estimates of the consequences of specific policy choices. Confidence that a policy significantly reduces the risk of infection may be useful even in the absence of a reliable estimate of absolute risk.

We begin with some preliminary definitions, then describe the model's assumptions, before describing the model proper.

### 2.0.1 Preliminary definitions

All the terms we define in this paper are included in Appendix 6. For now, we need the following terms:

By an *activity* we mean a well-defined set of interactions with clear bounds taking place over a period of time less than a day, for example a trip to a grocery store, or taking an airplane flight, or attending a sporting event as a spectator.

By the *participant* we mean a person attending the activity, whose probability of becoming infected we wish to model.

A *neighbor* at an activity is a person not in the participant's immediate household who, for some part of the activity, is close enough to pose a risk of air-borne infection. We shall say the neighbor is in the participant's *vicinity* if they are close enough to be a risk of infection. The precise nature of the vicinity is currently unknown; the CDC asserts that most infections are caused by individuals within 6 feet of each other [10], so a 6 foot radius may be an approximation for vicinity.

### 2.0.2 Assumptions

Like any model, there are assumptions about the state of the world that are necessary for the model to apply. We state them here with some explanation, and discuss them in more detail in Section 4:

A1 Our first assumption is that the probability of infection is additive over sub-activities. This means that if one segments the activity into sub-activities, the probability of getting infected over the whole activity approximately equals the sum of the probabilities over each segment.

Mathematically, this says that if we break an activity $A$ up into $N$ distinct sub-activities, $S_1, \ldots, S_N$ say, then the probability $p$ of becoming infected during activity $A$ satisfies

$$p \approx x_1 + \ldots + x_N, \tag{1}$$

where $x_j$ is the probability of becoming infected during $S_j$.

We cannot actually expect exact equality in (1). Nonetheless—and this is an essential point—we can reasonably expect that the left-hand side and right-hand side of (1) agree with each other to within 10% or less. We give a mathematical proof of this assertion in Appendix 7.

A2 For each sub-activity $S_j$ the probability $x_j$ of infection is the sum over the forms of transmission of independent probabilities, each of which has a multiplicative form.

A3 If a neighbor is not infectious, there is 0 risk of infection from them.

A4 If a neighbor is infectious, the probability that they will infect the participant depends on the distance away, whether they are facing towards or away from the participant, mask usage, viral load in the neighbor, sneeze etiquette, air circulation, and other factors.

A5 The probability of infection from a neighbor is linear in the amount of time spent in their vicinity. See Appendix 7 for a justification of this assumption.

A6 The probability of infection in each segment is independent of the other segments.

Formally, the model does not need the following assumption, but it will be important when applying the model:

A7 There is no increased chance of infection from members of the participant's immediate household engaging in the same activity, and we will ignore transmission from one's immediate household members. (For example, sitting beside a household member at an activity will be treated as zero-risk).

**2.1 Additivity over time.**   Assumption A1 is crucial to our study. Suppose we know an upper bound $v$ on the chance of an individual becoming infected by SARS-CoV-2 over the course of a day's activities. For some given activity $A$, such as attending a sporting event or taking an airplane flight, we break the activity up into temporally disjoint sub-activities, $S_1, \ldots, S_N$. (For example: entering the stadium, walking to one's seat, sitting and watching the event, going to a restroom, leaving the stadium). Suppose the probability of becoming infected in each subactivity $S_j$ is $x_j$, and we wish to estimate $p$, the probability of becoming infected at some time during $A$. In Appendix 7 we prove that $s = \sum_{j=1}^{N} x_j$ is a good approximation to $p$ using assumption A6 to do so, and the smaller $v$ is, the better the approximation. In particular, we show:

**Theorem.** *The following inequalities hold*:

$$
\begin{aligned}
0.95\, s \ \leq \ p \ \leq s \qquad & \text{if } v \leq 0.10 \\
0.90\, s \ \leq \ p \ \leq s \qquad & \text{if } v \leq 0.20 \\
0.75\, s \ \leq \ p \ \leq s \qquad & \text{if } v \leq 0.46.
\end{aligned}
$$

In Appendix 7.2, we derive some estimates for $v$ based on previous studies. For example, using numbers from an analysis of the Diamond Princess cruise ship [11], we calculate that $v$ is less-than-or-equal to 0.18, while combining US data from [12] with epidemiological modeling from [13] and [14] leads to an estimate that $v$ is less-than-or-equal to 0.37.

A simple example is useful to get some intuition about assumption A1 and the theorem. Suppose we were interested in the probability of rolling at least a single six when we roll three dice. That probability $p$ is simple to calculate: it is $p = 1 - \left(\frac{5}{6}\right)^3 \approx 0.42$, because the rolls are independent and the probability of rolling anything other than a six for a single dice is $\frac{5}{6}$. However, notice that, for this example, $x_j = \frac{1}{6}$ for all $j \in \{1, 2, 3\}$, and thus $s = \sum_{j=1}^{2} x_j = 3 \cdot \left(\frac{1}{6}\right) = 0.5$. If we were not able to calculate $p$ exactly, $s$ is an approximation of $p$ that is within 0.1 of the true probability. However, notice that, as the number of dice increases, $s$ will quickly approach—and then equal, and then exceed—unity, despite a roll of six never being guaranteed.

A1 is crucial because, if we assume that

$$p \approx x_1 + \ldots x_n,$$

then it follows that:

- A given absolute reduction of risk in any segment $S_j$ has approximately the same overall impact on $p$.

- One can compare the relative risks from different activities, such as going grocery shopping, flying, or attending a sporting event, by analyzing sub-activities.

## 2.2 The full model

Putting together all of our assumptions, we wish to model the probability that a participant at an activity contracts SARS-CoV-2. Actual infection is understood to happen in one of three ways [15]:

- Airborne transmission from an infectious neighbor at the activity, either by droplets or aerosols.

- Touching a contaminated surface, and then touching the participant's face before thoroughly washing the hands.

- Direct physical contact with an infectious person.

Each activity $A$ is broken down into a sequence of segments $S_j$, $j \in \{1, \ldots, N\}$, disjoint sub-activities each of which can be thought of as a single uniform event, either as a single event (e.g. going to the restroom) or an event with constant parameters (e.g. sitting for some period of time with one neighbor 3 feet away, 2 neighbors 6 feet away, and no other neighbors within 10 feet).

Following A3, A4, and A5, for each segment $S_j$, the probability that the participant becomes infected by air-borne transmission is the sum over every neighbor of [the probability the neighbor is infected] times [the probability the neighbor will cause the participant to be infected per unit time] times [the time spent in their vicinity]:

$$x_j^A = \sum_{n \text{ is a neighbor}} \tau_{j,n}^A \Pr[n \text{ is infected}][\text{time of } S_j].$$

Here $\tau_{j,n}^A$ is the probability per unit time that given the configuration (distance away, orientation, mask-wearing or not, etc.) that if neighbor $n$ is infected, they will infect the participant by air-borne transmission.

Similarly, the probability that the participant becomes infected by surface-born transmission from a surface they touch is [probability the surface is contaminated] times [probability they touch their face before washing their hands] times [probability that the touching leads to an infection]:

$$x_j^S = \sum_{\text{Surfaces}} \tau_j^S \Pr[\text{surface is contaminated}],$$

where $\tau_j^S$ is the probability that the participant will convey the infection from the surface to themselves.

Finally, the probability that the participant becomes infected by direct contact with an infected neighbor is [probability neighbor is infected] times [probability of touching] times [probability of transmission]:

$$x_j^D = \sum_{n \text{ is a neighbor}} \tau_{j,n}^D \Pr[n \text{ is infected}]\Pr[\text{touch } n],$$

where $\tau_j^D$ is the probability that if $n$ is infected and the participant touches $n$, then infection will be transmitted.

Combining the above, we have (using A1, A2, and A6):

$$x_j = x_j^A + x_j^S + x_j^D,$$

and

$$p = \sum_{j=1}^{N} x_j.$$

**Our contention is that this model is strategically valuable even without knowledge of the parameters** $\tau_{j,n}^A, \tau_j^S, \tau_{j,n}^D$. Even with no or highly uncertain knowledge of their values, the model allows one to estimate which activities (and which sub-activities) pose the most risk of infection. As we argue in Section 4, combined with knowledge of costs and benefits, that would allow for policymaking to decide on a set of activities to allow, and where adjustments can be made to lower the risk of those activities (e.g., choosing among competing seating configurations for an event venue).

In the next section, we outline how the model could be applied to idealized versions of airplane travel, attending a sporting event, sitting in a classroom, going to a restaurant, and attending a religious service. These examples are meant to give an idea of the value of knowing relative risks, but they do not contain the kind of fine-grained detail known by subject-matter experts that would be required for anything more than a first-order analysis.

## 3 Example calculations

We begin with an outline of how one could apply the model to commercial air travel. We then sketch the analysis for attending a sporting event, sitting in a classroom, going to a restaurant, and attending a religious service. An Excel program that allows one to see all the calculations and change the value of the inputs is available in the online Supplemental Information for all but the sporting event example (see Data Sharing Agreement).

### 3.1 Example: Air travel

Let us take travel on an airplane as an *activity*, as defined in Appendix 6. To use the model, we need to enumerate sub-activities $S_j$ that together make up the air travel activity, $A$. The sub-activities $S_j$ are:

1. boarding the plane

2. moving to and entering one's seat

3. sitting on the plane for the duration of the flight

4. leaving one's seat, and deboarding the plane.

   The relevant parameters for this question, with sample values which can be changed, are:

1. *Position of seats in the plane.* We employ the seating arrangement used by United Airlines for the Boeing 737, which is available on United's website [16]. Fig 1 shows the seating plan, with 50% occupancy.

2. *Seating arrangement.* This can differ between scenarios, but for this example we will assume the plane is full.

3. *Time spent boarding, and traveling to one's seat while on the plane.* We will assume the passengerstands in line for 10 minutes boarding, and takes 20 seconds to sit down at their seat

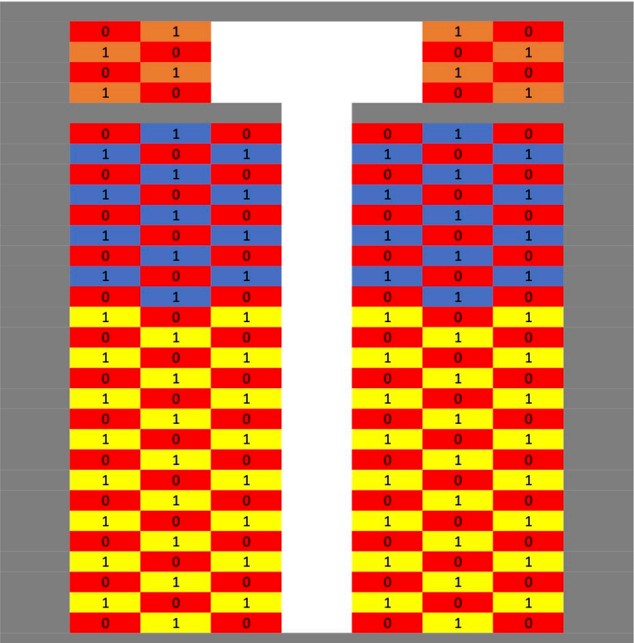

**Fig 1. An example seating arrangement of a Boeing 737 at 50% capacity.** Red cells (that are also labeled with 0s) are empty seats. Orange cells represent business class, blue cells represent premium economy, and yellow economy class. Cells with 1s are occupied.

once they reach the correct row on the plane. These values are estimates, and will vary according to airline boarding protocols.

4. *Order of seating*. We shall assume that the plane fills back to front, so that while walking to one's seat, one does not pass already seated passengers.

5. *The distance apart people stand while boarding*. This can vary based on preventative measures taken by airlines; for a first approximation, we will use 1.5 foot spacing.

6. *Duration of the flight*. This example will take flight duration as 180 minutes.

7. *Deboarding*, which will be modeled the same way as boarding for this example.

8. *How risk decays with distance*. There is much discrepancy in the literature as to this decay [5, 17]. Let us assume risk is inversely proportional to the square of distance from the source.

We shall also assume for this example that there is no direct physical contact between participants and that all surfaces are disinfected.

For each sub-activity, a participant is exposed to some amount of risk from their neighbors. As we do not know absolute risks, we will quantify the risks of the various sub-activities using the hazard × exposure model described in the previous section. Given the analysis is one of *relative* risk, we do not have an absolute unit to use in the quantification of risk; thus, as a basic risk unit, we will use the risk of spending one minute at a distance of one foot from a stranger. Appendix 9 and the online Supplemental Information contains more detail on the calculations presented below.

**3.1.1 Boarding and deboarding.** While boarding and deboarding, some number of strangers in the vicinity contribute to the risk a participant incurs. We assume that the

boarding process arranges passengers linearly, and that the risk posed by strangers further than 6 feet away is negligible. The risk for boarding is obtained by summing the risk contribution of two strangers each 1.5 feet away, two strangers each 3 feet away, two strangers each 4.5 feet away, and two each 6 feet away for a duration of 10 minutes. Note that if parameter #5 above—separation distance—were to change, the number of strangers for whom a risk contribution is calculated would also change.

Quantitatively, the risk is

$$10 \times \left( \frac{2}{(1.5 \times 1)^2} + \frac{2}{(1.5 \times 2)^2} + \frac{2}{(1.5 \times 3)^2} + \frac{2}{(1.5 \times 4)^2} \right) \approx 12.7.$$

**3.1.2 Entering and exiting seats.** Entering and exiting seats and sitting on the plane are calculated similarly to each other, but rather differently from boarding and deboarding. While entering or exiting seats, on average, we calculate the risk contribution of each surrounding seat, sum them, and multiply by the duration taken to be 0.33 minutes. This works out to 1.1 for entering and for exiting. The cumulative risk for boarding, sitting, leaving the seat, and deboarding is thus $2 \times (1.1 + 12.7) \approx 27.6$.

**3.1.3 Risk while seated.** The risk calculation for sitting on the plane must account for the fact that some seats are spaced less densely on the plane than other seats. The methodology here is to calculate the average risk a participant incurs from their neighbors while seated. This value will not be the risk any individual passenger actually incurs, but is more accurate for the plane as a whole. The average risk value per minute is 1.84, so the average risk from sitting on plane for a three hour flight is 331.0. Thus, the average risk a participant incurs for this activity is $331.0 + 27.6 \approx 359$.

**3.1.4 Changing parameters.** Given the varying practices of the major airlines [18], the percentage of occupied seats is one parameter for which we are already seeing wide variation The results for similar scenarios are:

Case 1 Airplane full, 1.5ft distancing while boarding
    Risk: 359

Case 2 Middle seats empty, 3ft distancing while boarding
    Risk: 146

Case 3 Airplane half full, 6ft distancing while boarding
    Risk: 100

Although the numbers 359, 146 and 100 are not in absolute units, they do show the relative effect of different possible mitigation strategies. For example, removing roughly one-third of the passengers by keeping the middle seats empty and increasing social distancing while boarding (Case 2) more than halves the risk compared to the full airplane. Other scenarios could be modeled similarly.

In each of the above cases, we used the inverse square decay function to model the change in risk as distance to others changes. While the other parameters used in the model are measurable, the rate of decay of risk with distance has not been experimentally verified, and is quite uncertain. To see how sensitive to the decay function our conclusions are, we will try two very different decay functions of risk with distance. The first has a very slow exponential decay [5], and was based on averaging over many different studies; it concluded that each additional meter of distance decreased risk by a factor of 2.02. The second has a very rapid decay, based on simulations of droplet dispersion [17]. We shall refer to these as the Chu model and the Chen model, respectively. To normalize, we multiply the output of each function by a constant

**Table 1. Relative risks with different decay assumptions for a three-hour flight.**

|  | Case 1 | Case 2 | Case 3 |
|---|---|---|---|
| Inverse Square | 359 | 146 | 100 |
| Chu Model | 359 | 230 | 170 |
| Chen Model | 359 | 64 | 35 |

such that the risk for a full flight of three hours is the same for each risk decay model. The risk values of each case with each other decay model are listed below in Table 1.

The relative risk at 2 meters compared to 1 meter is 25% in the inverse square model, 49.5% in the Chu model, and 4.2% in the Chen model.

Despite the great differences between the three model's decay rates, there are many similarities in the relative risks for the three scenarios. In all cases, the vast majority of risk is incurred while sitting, and Case 3 is about 2/3 the risk of Case 2 no matter the decay assumption. However, in the Chen model, the risk for Case 2 relative to Case 1 is much more reduced than it is under either of the other two decay assumptions. Fig 2 summarizes how the risk score changes as a function of flight time, the cases considered above, and the decay model assumption. The values at $t = 0$ show the contributions of the boarding/deboarding risk to each scenario, and the relatively larger risk in the Chen and Inverse Square models compared to the Chu model. The risk scores are normalized so that the risk at the three-hour mark is the same across decay models, revealing that, under the Chu model, removing passengers from the flight (Cases 2 and 3) has less of an effect than under the other two models.

The very large discrepancy in the relative risk in Case 1 to Case 2 is explained by the extreme sensitivity of the Chen model at close ranges. According to this model, moving from 0.2 meters to 0.3 meters reduces risk by a factor of 88. If a sensitive model such as the one presented by Chen et al. is most accurate, it will be important to be aware of the interval or intervals where risk drops rapidly.

Using the idealized example above of the airplane analysis, one can see the relative benefits of different mitigation strategies. Given the contribution of the time spent seated to the total risk score and what is currently known about mask-wearing, making masks mandatory could be a (cost-)effective strategy [19, 20]. Our analysis shows that keeping the middle seat vacant unless there is a party of three travelling together at least halves the risk, under a very wide range of decay assumptions. Managing boarding is likely less costly than leaving seats empty, but our analysis finds that the total impact will be lower that adjustments to the seating plan because it takes up a small part of the total flight time. Ongoing research will likely lead to increased knowledge about mitigation strategies specific to air travel that could be effective [21].

In addition to considering variation in parameters, there are other structural elements of the scenario that could affect the analysis. For example, it is possible that passenger compliance could vary. The above analyses have assumed that passengers follow a sequence of steps, such as boarding the plane a certain way, wearing a mask for nearly the entire flight, etc. [22]. The extent to which people comply with specific norms about protective actions varies by group and by hazard [23]. Social norms, social influence, and social comparisons all play a role in determining what people will do [24–26]. Planning for aberrant responses to airline (or other) policies about protective actions should be part of any private or public entity's analyses of the possible outcomes of activities during the pandemic. Incorporating uncooperative members of the public into an analysis could demonstrate their possible negative effects, bolstering arguments for policymaking to mitigate those effects (e.g., by empowering staff with the regulatory powers to deny services to such persons when they do not have such powers already).

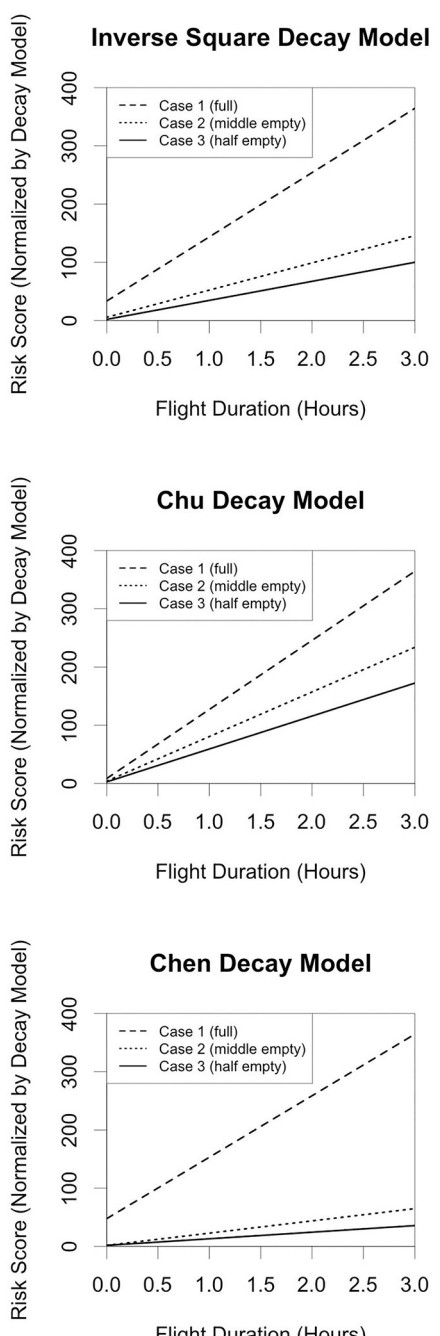

**Fig 2. Risk score as a function of flight duration, decay model, and seating/social distancing plan.** The three plots show how the risk score changes as a function of flight duration, decay model (inverse square, Chen, and Chu models), as well as seating plans with social distancing assumptions that are described in the main text. Note that the *y*-axis in each plot is a normalized score. As a result, absolute values cannot be compared between plots, but other characteristics can (e.g., rate-of-change, difference between scenarios etc.).

## 3.2 Example: Attending a sporting event at a stadium

We base our analysis on the TD Garden Stadium in Boston, MA, in the United States.

Here, the relevant sub-activities are

1. entering the stadium

2. moving to and entering one's seat

3. sitting in one's seat

4. getting food and or drink at a concession stand

5. eating in one's seat

6. going to the bathroom

7. leaving the stadium

The relevant parameters for this question are:

1. *Mask protocol.* We assume masks are required except when eating.

2. *Seating arrangement.* For different scenarios, one can map the stadium, with certain seats kept empty, and others sold in small groups to trusted cohorts. By a "trusted cohort", we mean somebody like a household member with whom one spends so much time in close contact that their presence at the event does not constitute an added risk. This is significant, because having multiple members of the same cohort sit together, while other cohorts sitting at a distance, means that there will be little droplet risk (though long-range aerosol transmission between different cohorts must still be considerd. For each person in the stadium, the distance from their seat to all other seats occupied by strangers (up to some cut-off distance, say 6 feet) is known.

3. *Time spent entering.* This depends on the number of entrance turnstiles, what the protocol is, and whether arrival times are deliberately staggered. We will assume 0.25 minutes.

4. *Time spent walking to one's seat.* This depends on the lay-out of the stadium, walking speed, and density of people. Assuming staggered entrances, we estimate this at 8.3 minutes.

5. *Social distancing requirement in corridors.* This can be set by policy; we assume 3 feet.

6. *Duration of game.* Assume 190 minutes.

7. *Concessions.* This has two parts: ordering and eating. The latter is much more significant, as without a mask, and with the potential for one's food to be contaminated, the risk of both infecting and being infected during eating is much higher. We will assume that a person eats for 15 minutes, that they are 4 times more likely to transmit infection without their mask, and that while eating they are 3.5 times as likely to be infected. (The factors of 4 and 3.5 are just guesses; as research is conducted, more accurate figures can be substituted).

8. *How risk decays with distance.* Let us assume risk is inversely proportional to the square of distance from the source.

9. *Aerosol risk.* One of the thorniest questions in devising a relative risk model is how to account for both aerosol and droplet risk. The relative risk of long-range aerosol transmission compared to short-range transmission from immediate neighbors is not known, though considered significant [6]. There are calculators that estimate the aerosol risk, such as the CU Boulder COVID-19 Aerosol transmitter tool [27]. However, this tool explicitly excludes droplet transmission, and assumes that 6 foot social distancing is always maintained. Despite increasing awareness of the significance of aerosol transmission, there are no published studies that numerically compare aerosol risk to droplet risk. So to make a model that incorporates both, we have to make some assumption about their relative risks.

This assumption can of course be changed as more information emerges.

In the paper [28], it is argued that the aerosol risk is approximately $c/V$, where $V$ is the the ventilation, measured in cubic foot per minute per person, and $c$ is an empirical constant. We will set $c = 1$, meaning the droplet risk at 1 foot is the same as the aerosol risk if the ventilation is 1 cubic foot per person per minute.

10. *Air volume of the stadium.* If the stadium starts out empty and clean, the aerosol risk will be reduced somewhat.

11. *Bathroom design and constraints.* The assumptions we make, based on TD Garden stadium, are that at most 4 people will be in the bathroom at any one time, with an average visit of 4 minutes. The ventilation in the bathrooms is 1075 cfm, which is 127 liters/person/second. This very high ventilation rate leads to a very low aerosol risk. Every other sink and urinal will be blocked off, leaving a gap of 6 feet between users. There will be some passing time as people enter and exit; we estimate an upper bound of 1 minute at one foot, which is the large majority of the risk estimate of going to the bathroom.

12. *Presence or absence of screening of attendees, that would catch some percentage of infectious people.* For this model, we will assume it is not available.

Using these parameters (included in our supplemental material), we arrive at the following scores:

- Full stadium: 1044 risk units, 696 of which come from the seated portion.

- Half-full stadium: 335 risk units, 219 of which come from the seated portion.

- 21%-full stadium: 125 risk units, 77 of which come from the seated portion.

- 21%-full stadium, no eating or drinking: 83 risk units.

Based on these calculations, concessions pose a larger risk the fuller the stadium. However, many of the relevant parameters are uncertain or unknown, but with better estimates the above set of steps could be used to provide more realistic estimates.

## 3.3 Example: Classroom

Consider a classroom, with one possible seating arrangement as in Fig 3, where 1 represents an occupied seat, and 0 an empty seat. One of us (JEM) went to a classroom at the Washington University in St. Louis and measured its layout: seats are 3.9 feet apart horizontally, and 4.3 feet apart vertically. For that classroom, its volume is approximately 5834 cubic feet, and the

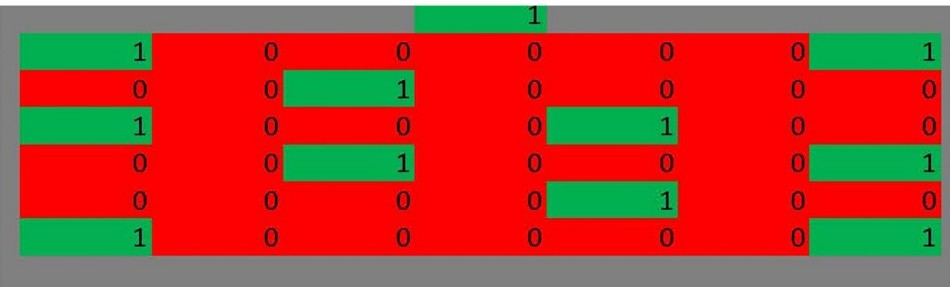

**Fig 3. An example classroom layout.** Red cells (that also contain 0s) indicate empty desks, while green cells (that also contain 1s) indicate occupied desks.

ventilation is 8 liters per person per second when full to capacity of 43, which is 729 cubic feet per minute.

Let us assume that a class is 60 minutes long, and that the time taken to reach seats is negligible. Moreover, assume that everybody is required to wear a mask at all times. As before, we shall set $c = 1$, and assume an inverse square decay of droplet risk with distance, and that the droplet risk becomes to 0 at 6 feet.

Using these parameters, there are two different risk scores. One is the steady state score, whereby one assumes that the rate of exhalation equals the amount of exhaled breath removed by the ventilation system. The other takes into account that if the classroom starts out clean and empty, it takes a while for the air in the classroom to reach this equilibrium state. If $T$ is the total duration (here, 60 minutes) and $f$ is the fraction of air in the room removed per minute (here, 0.125), this correction factor reduces the effective time in the room for aerosol exposure (but not droplet exposure) to

$$T + \frac{1}{f}\left(e^{-f*T} - 1\right)$$

which in this case is 52 minutes. Table 2 shows the two risk scores given different occupancy levels and configurations, showing a surprisingly slow decay of risk with number of people—it is not much better than linear. The Excel program in the online Supplemental Information contains the calculations and allows the interested reader to adjust the parameters and see the change in risk scores.

## 3.4 Further examples

Using similar procedures, one can analyze other events. More detail is provided in the Excel program provided in the online Supplemental Information.

**3.4.1 Restaurants.** Assuming that the key variables here are seating arrangement, which tables are occupied, ventilation rate, time spent at table, how long the wait is to get in, and how crowded the waiting area is. The ratio of aerosol to droplet risk must still be estimated. The principal difference between indoor and outdoor seating is that, under current understanding, there is very little aerosol risk when people are separated outdoors. When indoors, the aerosol risk depends on the ventilation (and whether the air is being replaced by fresh air, or cleaned by a virus removng filter, or just being recirculated).

For one sample restaurant we modeled, we calculated a risk score for a 60 minute meal, assuming everybody at each table is in the same trusted cohort (so they could only be infected from other tables) to be 87 if the restaurant is full, 41 if it is half-full. Of this, the aerosol risk was estimated at 13 and 6, respectively.

**3.4.2 Religious service attendance.** This situation can be modeled similarly to a classroom. The spacing of seats in the church must be determined on a case-by-case basis. Our analysis yielded a risk score of 140 risk units when full, 69 when half-full, for a one hour

**Table 2. Relative risks with different seating assumptions.**

| Number in room | Configuration | Clean Room | Steady State |
|---:|:---:|:---:|:---:|
| 11 | As in Fig 2 | 23 | 30 |
| 22 | Checkerboard | 62 | 68 |
| 22 | Every other row | 66 | 72 |
| 25 | Every other column | 63 | 69 |
| 43 | Full | 121 | 125 |

service. These scores decrease if people go in trusted cohorts (more likely at a religious service than in classrooms).

### 3.5 Summary

In this section, we have endeavoured to outline how one might use the approach described in Section 2.2 to frame an analysis of the risk of SARS-CoV-2 infection during a variety of activities. Of course, decision-makers and analysts in any of the above public or private decisions would need to incorporate more sensitivity analyses and more details specific to those contexts, details that would be known to subject-matter experts. We believe that, when the relevant assumptions hold, the approach we have outlined can be useful in identifying the most dangerous sub-activities of an activity, which could be used to inform policy decisions including strategies to mitigate the risk posed by those sub-activities. In the next section, we discuss the advantages of the approach in more detail, as well as its limitations and the future work that would be required for a responsible application of the approach by policymakers.

## 4 Discussion

In this paper, we have described a model for estimating the relative risk of infection by SARS-CoV-2 during an activity that lasts less than a day. Even without being able to estimate absolute risks, it allows decision-makers to rank activities according to how much risk of infection they pose to the public; in combination with knowledge about those activities' costs and benefits, decision-makers can make more informed choices about whether, and how, to allow people to participate in currently forbidden activities [29]. Crucially, as illustrated in the example calculations, an analysis using this model also reveals which segments of the activity pose the greatest risk. When these are modifiable, stakeholders can act to lower the risk.

In contexts where the model applies, it has significant policy value, despite only being able to calculate relative risks:

- The model is linear in time. Therefore, engaging in an activity for twice as long doubles the chance that a participant becomes infected. Boarding airplanes, for example, can be done in much more efficient ways than is currently the norm [30]. Optimizing the boarding process so that passengers spend less time close to neighbors will reduce their infection risks.

- The model is linear in the proportion of attendees at an activity who are infectious. This number in turn is the product of two numbers: the proportion of the population who are infectious (which will vary over time) and the probability that an infectious person will not self-isolate. The latter number can be reduced by public health education, by testing and contact tracing, and by health checks.

- The model is linear in the probability that a given neighbor will cause an infection. In [31] the authors find that home-made masks block 95% of air-borne viruses, medical masks block 97%, and N95 masks block 99.98%. That study was a mechanical simulation using nebulizers and did not distinguish in application between a potentially infectious person wearing a mask to reduce their probability of transmission, and an uninfected person wearing a mask to reduce their probability of infection. In [5], the authors perform a meta-analysis, and find that wearing masks reduced risk, with high uncertainty in the amount, but their point estimate was a reduction to 23% of the non-mask wearing risk when using non-respirator masks (and a reduction to 4% using respirators). In [32], the effectiveness of masks in practice is considered.

- The model is linear in density of neighbors. Reducing the number of neighbors at a given distance by 50% reduces by 50% the chance of air-borne infection. Leaving seats open, and clustering only members of the same household, will reduce the risk of air-borne transmission.

- For surface-borne infections, the model is linear in the probability that the surface is contaminated. Doubling the cleaning frequency will approximately halve the probability that the surface is contaminated. Indeed, if there is some small constant rate $\varepsilon$ at which the surface becomes contaminated, the probability it is infected at time $t$ after cleaning is $\varepsilon t$. So if it is cleaned every $T$ minutes, and someone touches it at a random time between 0 and $T$, the probability they are touching an infected surface is

$$\frac{1}{T} \int_0^T \varepsilon t \, dt \;=\; \frac{\varepsilon T}{2}. \tag{2}$$

Halving $T$ halves the right-hand side of (2).

These linearities allow for comparisons among different scenarios, and comparisons across different activities.

Of course, deciding whether and how to relax restrictions on activities requires understanding the public's perceptions of the risks, costs, and benefits of doing so. Formally equivalent risks could be perceived differently, in ways that might seem irrelevant to a risk analyst but would impact the decisions of potential participants in an activity [33]. The mental models approach to risk communication [34] would provide one way in which to systematically study those differences, by determining how the public views the risks of COVID-19 differently than the experts performing the risk analysis—are the parameters and their relationships, the possible outcomes, and the probabilities of those outcomes viewed similarly? Approaches inspired by the psychometric tradition could place COVID-related risks on the "dread-unknown" dimensions of risk that have been found to effectively characterize a wide-range of risks faced by the public [35, 36]. Doing so could help decision-makers understand how COVID-related risks compare to other risks, and design their reopening strategies informed by similarities and differences in perceptions of those less-novel risks, including what is known about the public's actions about them (e.g., commercial air travel and the risk of terrorism). Studies that characterize the public's risk perceptions will be essential to ensuring that the analysts' assumptions about the public's understanding of the activity are empirically grounded, and account for any differences (e.g., by informing the design of risk communications meant for public consumption) [37].

Furthermore, so long as there is some nontrivial amount of virus in the community, activities involving large numbers of people will almost certainly lead to some eventual transmission. Decision-makers need to evaluate the testing and contact-tracing infrastructure of the jurisdictions where the activities are located to determine whether that transmission can be contained, given that any transmission due to the activity is a burden not only to the participants, but also to the entire community. Decision-makers need to invest in empirically-tested risk communication so that participants understand the risks accurately and the public at large understands why that risk is judged to be acceptable by policymakers. There are already detailed frameworks for engaging the public about scientific and technical risks and undertaking an analytical-deliberative process to develop a plan that is widely endorsed among stakeholders [38]. There is no reason these frameworks cannot be adapted for the SARS-CoV-2 pandemic [39].

The model should be conceived as a tool in a bottom-up-analysis—there is no one-size-fits-all approach to the problem. Each activity has specific characteristics only known to those involved with the activity; even enumerating them can require specialist knowledge. We have attempted to outline how the model might work in a variety of scenarios, but none of those examples has the level of detail required to be used "off-the-shelf," especially given that some of the examples used placeholders for parameter estimates that would be required for an actual implementation. The June 2020 report describing how the entertainment industry can safely return to work [40], jointly authored by the major unions of that industry, is an example of the synthesis of modeling, industry knowledge, and risk communication that is possible (and required) when an interdisciplinary team that includes experts about the specific industry creates COVID-related policies for that industry.

### 4.1 Limitations

Like any model, the model described here depends on its assumptions. In our view, the most problematic assumptions are A2 and A6, which require independence. That could fail, if, for example, infection requires a certain minimum threshold of exposure. Similarly, A7, the assumption that those in one's household pose no threat, could be problematic. The presence of family members could increase the risk of exposure after the activity when one returns home, given, e.g., their separate trips to the washroom or through the turnstile during the activity. The importance of these extra exposures is an empirical question, and a function of the relative risk of those sub-activities done individually, and protective actions taken after the event (e.g., social distancing, hand-hygiene, proactive testing etc.).

As it stands, our model cannot account for household transmission, given that "living in a household" violates our definition of an "activity" because living in a household lasts much longer than a day, by its nature. Furthermore, decomposing (a day in) a household into its constituent subactivities would be difficult given the variation and number of subactivities. In contrast, when attending a sporting event, there are only so many things one can do in the stadium, and, in fact, subactivities can be constrained by those running the activity in order to lower the risk of infection. That kind of control over subactivities is unlikely to be generalizable to households.

Household transmission has been found to be an important mode of transmission [41], but there are strategies to mitigate it [42]. If people begin to partake in more activities outside of the home, finding ways to encourage increased protective actions within the household prophylactically would help counteract the additional risk posed by leaving one's home.

More importantly, the model assumes that the risk of infection is a function of the background risk in the population of the activity's jurisdiction. However, that assumes the subpopulation of potential participants does not have more virus prevalence than the community at large. Whether that is true is also an empirical question. For example, are those who would choose to attend a stadium concert during a pandemic more or less likely to participate in protective actions that lower their overall risk of virus infection or of virus transmission? Part of the empirical study of the public's risk perceptions would need to include an appraisal of that question.

Furthermore, the model has a specific definition of "activity," and it is crucial that the definition is clear to those who would use the model. For instance, considering a semester on a university campus as ∼90 separate one-day activities would not be an appropriate use of the model, because the population on campus from one day to the next is almost identical. Unless participation in an activity incurs no additional risk for a participant beyond their usual activities, decision-makers would need to consider how to limit individual participation; for example, in limiting the number of sports games one can attend in a given time period. Otherwise, the risk of transmission among participants will surely increase over the average risk in the

community as previously-shuttered activities become a part of their day-to-day life but not of the lives of their fellow community members. The feasibility of such controls should be considered during the decision-making process.

For analytical purposes, our examples show how sensitive many analyses will be to uncertainty about how the probability of infection decays with distance, including the threat of long-range exposure [5, 17, 43]. As long as estimates for those values vary widely in the literature, decision-makers may need to be conservative in cases where an analysis is highly sensitive to changes in the relevant parameters.

Finally, of the various modes of transmission—droplet, aerosol, fomite, and direct contact—it is not known how they compare to each other. Even for the two airborne modes, droplets and aerosols, there is disagreement about their relative importance [3, 6, 15, 44, 45]. However, the model is adaptable enough that different estimates of the relative importance of these pathways can be included, and the model can use whatever assumptions the scientific evidence supports best at the time of use.

## 5 Conclusion

The SARS-CoV-2 pandemic has restricted the activities of every person in the world. As governments and businesses try to decide how to reopen society, they need an analytical framework with which to make reasoned decisions. While much of the modeling done to date has focused on estimating the parameters needed to calculate the absolute risk of SARS-CoV-2 infection, here, we focus on estimates of relative risk. Such a model should allow decision-makers to rank the risk of activities and their constituent discrete sub-activities. Combined with an accounting of their benefits and costs, decision-makers could make better-informed decisions about those activities. At the time of writing, vaccines for COVID-19 have become available for a small proportion of the population. As those vaccines become more widely available, our approach could be modified to account for estimates of the percent vaccinated of the population of potential participants in an activity.

## 6 Appendix: Definitions

By an *activity* we mean a well-defined set of interactions with clear bounds taking place over a period of time less than a day, for example a trip to a grocery store, or taking an airplane flight, or attending a sporting event as a spectator.

By the *participant* we mean a currently non-infected person attending the activity, whose probability of becoming infected we wish to model.

A *neighbor* at an activity is a person not in the participant's immediate household who, for some part of the activity, is close enough to pose a risk of air-borne infection. We shall say the participant is in the neighbor's *vicinity* if they are close enough to become infected.

$v$ is the probability that withour social distancing, an infectious individual would infect at least one susceptible individual over the course of one day.

$v'$ is the expected number of new infections per day caused by an infectious individual without social distancing.

$\tau$ is the doubling time of the infection (see below).

## 7 Appendix: Additivity in time

### 7.1 Mathematical derivation of approximate additivity

Let us assume that an activity $A$ is decomposed into $N$ segments, called $S_1, \ldots, S_N$, and each segment $S_j$ has some risk $x_j$ of causing infection. We further assume that these risks are

statistically independent of each other (assumption A6). Then the probability $p$ of being infected at some time during $A$ is

$$p = 1 - \prod_{j=1}^{N}(1 - x_j). \tag{3}$$

Let

$$s = \sum_{j=1}^{N} x_j \tag{4}$$

Then we claim that

$$p \approx s,$$

where the symbol $\approx$ means "is approximately equal to". Indeed we claim that

$$\left(\frac{1 - e^{-s}}{s}\right) s \leq p \leq s. \tag{5}$$

To see (5), we use the arithmetic-geometric inequality to show

$$
\begin{aligned}
1 - p &= \prod_{j=1}^{N}(1 - x_j) \\
&\leq \left(1 - \frac{1}{N}\sum_{j=1}^{N} x_j\right)^{N} \\
&= \left(1 - \frac{s}{N}\right)^{N} \\
&\leq e^{-s},
\end{aligned}
$$

which yields the left-hand inequality in (5).

The right-hand inequality follows from

$$
\begin{aligned}
\ln(1 - p) &= \sum_{j=1}^{N} \ln(1 - x_j) \\
&= -\sum_{k=1}^{\infty}\frac{1}{k}\sum_{j=1}^{N} x_j^k \\
&\geq -\sum_{k=1}^{\infty}\frac{1}{k}\left(\sum_{j=1}^{N} x_j\right)^k \\
&= \ln(1 - s).
\end{aligned}
\tag{6}
$$

The correction factor between $s$ and $p$ is $f(s) = \frac{1 - e^{-s}}{s}$, in the sense that

$$f(s) \leq \frac{p}{s} \leq 1.$$

The function $f(s)$ is a decreasing function of $s$, and has a right-hand limit of 1 as $s$ tends to 0. Since $s \leq -\ln(1-p)$, we have

$$f(s) \geq f(-\ln(1-p)) = \frac{-p}{\ln(1-p)}.$$

So our conclusion is that we always have

$$\frac{p}{\ln(1/(1-p))} \leq f(s) \leq \frac{p}{s} \leq 1,$$

justifying the claim that $p \approx s$. For representative values of $f(s)$ and $g(p) = -p/\ln(1-p)$, see Tables 3 and 4. They can be read as saying that if we know the value in the top row is an upper bound on $s$ (respectively $p$) then the value in the second row is a lower bound on $\frac{p}{s}$.

## 7.2 Estimating an upper bound on $p$

As we saw in Subsection 7.1, how good the approximation $p \approx s$ is depends on how close $f(s)$ is to 1. How can we measure this?

We use the assumption that the activities we are considering last less than a day. While some activities are more risky than others, we further assume that all the events will be designed so that the total risk of an infected individual spreading the infection is no greater than it was at the beginning of the pandemic before any social distancing measures were in place. So we get an upper bound

$$p \leq v \tag{7}$$

where $v$ is the probability that before social distancing, an infectious individual would infect a susceptible individual over the course of one day. Note that for many activities, it is reasonable to assume that $p$ is much less than $v$, thus tightening the estimation in Subsection 7.1.

Since an infected individual can infect multiple susceptibles, the expected number they would infect over the course of the day would be slightly higher, namely $v' = \ln\left(\frac{1}{1-v}\right)$ if one assumes a Poisson distribution. (This is a small adjustment that will not materially affect our conclusion, so the reader can ignore it.)

Using a standard SIR model at the early stage of an infection, the proportion of the population that is susceptible is close to 1. So the proportion of the population that is infected will grow exponentially, like $e^{\kappa t}$ for some rate $\kappa$, where $1 + v' = e^{\kappa}$. The doubling time $\tau$ is the time at which $e^{\kappa \tau} = 2$, so $\kappa = \ln(2)/\tau$. Thus we get

$$v' = e^{\kappa} - 1 = e^{\ln 2/\tau} - 1$$

and

$$v = 1 - e^{-v'} = 1 - e^{1 - e^{\ln 2/\tau}}. \tag{8}$$

**Table 3.** $f(s) = \frac{1-e^{-s}}{s}$ as a function of $s$.

| $s$ | 0 | 0.05 | 0.1 | 0.15 | 0.2 | 0.3 | 0.4 | 0.5 |
|---|---|---|---|---|---|---|---|---|
| $f(s)$ | 1.00 | 0.98 | 0.95 | 0.93 | 0.91 | 0.86 | 0.82 | 0.79 |

**Table 4.** $g(p) = \frac{-p}{\ln(1-p)}$ as a function of $p$.

| $p$ | 0 | 0.05 | 0.1 | 0.15 | 0.2 | 0.3 | 0.4 | 0.5 |
|---|---|---|---|---|---|---|---|---|
| $g(p)$ | 1.00 | 0.97 | 0.95 | 0.92 | 0.90 | 0.84 | 0.78 | 0.72 |

What is $\tau$? In [46] they estimate that the doubling times in Chinese provinces in the period January 20—February 9 2020 ranged from 1.4 days (95% CI 1.2-2.0) in Hunan province to 3.1 days (95% CI 2.1-4.8) in Xinjiang province.

In [47], the authors estimate the doubling time in Italy in March 2020 to be 3.4, 5.1 and 9.6 days in the first, second and third ten day periods of the month.

In [11], the authors estimate the probability of infection in a crowded zone (summed over all neighbors in the vicinity) to be 1.8% per hour, and to be 0.18% and 0.018% in moderate and uncrowded zones, using data from the cruise ship Diamond Princess. If we assume at most 12 hours spent in crowded zones per day on the cruise ship, this would yield the estimate

$$v' \leq 1 - (1 - 0.018)^{12} = .20,$$

which in turn from (8) gives

$$v \leq 0.18.$$

In [13], the authors use an SEIR model on the data from [14], and take into account the incubation period (6 days) recovery period (14 days) and a mortality rate of 1%. They assume that transmission rates are the same in asymptomatic and symptomatic states, and get a value of $v$ that is 0.126. This follows from their equation

$$v = \frac{R_0}{\frac{1}{\delta} + \frac{1}{\omega^R + \omega^D}}$$

and using their values $R_0 = 2.5$, $\omega^R = 1/14$ is recovery rate from symptomatic to recovered, and $\omega^D = .01$ is the mortality rate.

The value $R_0$, the number of new people infected per infectious person, is widely reported by time and geographic region—see e.g. [12] for estimates of $R_0$ by U.S. state. If one makes the more conservative estimate that only asymptomatic carriers will be circulating, and using the same 6 day incubation period, then one gets the bound

$$v \leq \frac{R_0}{6}.$$

With an estimate of 2.22 for pre-mitigation $R_0$ in the U.S. [12], this gives the bound

$$v \leq \frac{2.22}{6} = 0.37.$$

Table 5 gives some values of $v$ as a function of $\tau$.

**Table 5. $v$ as a function of $\tau$, the doubling time.**

| $\tau$ | $v$ | $f(v)$ | $g(v)$ |
|---|---|---|---|
| 1 | 0.63 | 0.74 | 0.63 |
| 1.4 | 0.47 | 0.80 | 0.74 |
| 2 | 0.34 | 0.85 | 0.82 |
| 3 | 0.23 | 0.89 | 0.88 |
| 4 | 0.17 | 0.92 | 0.91 |
| 5 | 0.14 | 0.93 | 0.93 |
| 6 | 0.12 | 0.94 | 0.94 |
| 7 | 0.10 | 0.95 | 0.95 |
| 8 | 0.09 | 0.96 | 0.96 |

## 8 Appendix: Refinement to additivity over segments

One can refine the analysis in Appendix 7. Let us assume (3) and (4) both hold, and that we have segmented the activity into sufficiently small pieces that each $x_j \leq \varepsilon$ for some small $\varepsilon$ that we assume satisfies $0 < \varepsilon \leq \frac{1}{2}$.

Then we can tighten the bounds in (5) to

$$1 - e^{-s} \leq p \leq 1 - e^{-(1+\varepsilon)s}. \tag{9}$$

Indeed, to see this start with equality (6) to get

$$
\begin{aligned}
\ln(1-p) \quad &= \quad -\sum_{k=1}^{\infty} \frac{1}{k} \sum_{j=1}^{N} x_j^k \\
&\geq \quad -\sum_{k=1}^{\infty} \frac{1}{k} s\varepsilon^{k-1} \\
&\geq \quad -s - \varepsilon s \sum_{2}^{\infty} \frac{1}{k} \varepsilon^{k-2} \\
&\geq \quad -s - \varepsilon s \sum_{2}^{\infty} \frac{1}{k} \frac{1}{2^{k-2}} \\
&= \quad -s - \varepsilon s(-2 + 4\ln(2)) \\
&\geq \quad -s - \varepsilon s.
\end{aligned}
$$

Exponentiating, we get the right-hand inequality in (9).

## 9 Appendix: Airplane interior

We include further details on the airplane interior of the Boeing 737-800 from Section 3.1. Fig 1 in the main text shows the interior with 50% occupancy. In feet, the width of the seats is 1.71 in First class, and 1.4 in Economy Plus and Economy. The pitch is 3.08 in First, 2.83 in Economy Plus, and 2.5 in Economy. The aisle is 3.18 feet wide.

To calculate the average seating risk per minute for a given seating arrangement and risk decay, we calculate for each occupied seat the distance to each other occupied seat, and this gives a corresponding risk score based on the chosen decay model. These are all summed up, and then divided by the number of occupied seats, to get the average risk score per minute seated.

To calculate the risk of entering a seat, we assume the passenger spends a certain amount of time in the aisle using the overhead bin. In this example, we assume 20 seconds; of course, that could be changed. During that period, we calculate their distances from other passengers and then a corresponding risk score per minute.

## Supporting information

**S1 File.**
(XLSX)

## Author Contributions

**Conceptualization:** John E. McCarthy, Barry D. Dewitt, Bob A. Dumas, Myles T. McCarthy.

**Formal analysis:** John E. McCarthy, Barry D. Dewitt, Myles T. McCarthy.

**Methodology:** John E. McCarthy, Bob A. Dumas.

**Validation:** Myles T. McCarthy.

**Visualization:** Barry D. Dewitt.

**Writing – original draft:** John E. McCarthy, Barry D. Dewitt, Myles T. McCarthy.

**Writing – review & editing:** John E. McCarthy, Barry D. Dewitt.

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
