## [Decision Letter · Decision Letter 0]

24 Sep 2020

PONE-D-20-27466

A deterministic linear infection model to inform Risk-Cost-Benefit Analysis of activities during the SARS-CoV-2 pandemic

PLOS ONE

Dear Dr. McCarthy,

Thank you for submitting your manuscript to PLOS ONE. After careful consideration, we feel that it has merit but does not fully meet PLOS ONE’s publication criteria as it currently stands. Therefore, we invite you to submit a revised version of the manuscript that addresses the points raised during the review process. All reviewers expressed concerns that warrant a major revision.

We look forward to receiving your revised manuscript.

Kind regards,

Igor Linkov

Academic Editor

PLOS ONE

Journal Requirements:

"JEM, BAD and MTM received funding from Delaware North, a company that may be affected by the research reported in the paper.".

We note that one or more of the authors are employed by a commercial company: 'Omnium LLC'.

3. We note you have included  tables to which you do not refer in the text of your manuscript. Please ensure that you refer to Tables 1 and 4 in your text; if accepted, production will need this reference to link the reader to the Table.

Reviewers' comments:

Reviewer's Responses to Questions

**Comments to the Author**

1. Is the manuscript technically sound, and do the data support the conclusions?

Reviewer #1: Partly

Reviewer #2: No

Reviewer #3: No

2. Has the statistical analysis been performed appropriately and rigorously? 

Reviewer #1: Yes

Reviewer #2: Yes

Reviewer #3: No

3. Have the authors made all data underlying the findings in their manuscript fully available?

Reviewer #1: Yes

Reviewer #2: Yes

Reviewer #3: Yes

4. Is the manuscript presented in an intelligible fashion and written in standard English?

Reviewer #1: Yes

Reviewer #2: Yes

Reviewer #3: Yes

5. Review Comments to the Author

Reviewer #1: McCarthy et al. have formulated a “deterministic linear” modeling framework to inform risk-cost-benefit analysis of activities during the SARS-CoV-2 pandemic. The model outputs infection probabilities that are additive over defined sub-activities and infection pathways. Namely, the model assumes independence of the individual probabilities and that activities cannot last longer than one day. Then, the model is applied to estimate the risk associated with taking an airplane ride, under varying parameter choices. Furthermore, the authors argue that the linearity assumption makes the model easier to interpret, with minimal sacrifice in reliability.

This is a very timely topic and I would like to applaud the authors for tackling this topic using a more abstract, mathematical framework. The results suggest that the model has a high potential utility in estimating relative risks associated with various activities. I would recommend this paper for acceptance, but believe a couple aspects of the manuscript could be thoroughly revised. First, the paper needs further clarity in terms of definitions (i.e., “deterministic” model), assumptions, and presentation of the mathematics. Second, the model application to the airplane ride example should consider a wider range of parameters and augmented interpretation of estimated risk values. These revisions to the paper will enhance the demonstration of the model’s potential ability for application in risk estimation and decision making.

More detailed comments are provided in the attached document.

Reviewer #2: During the COVID-19 pandemic, difficult decisions are being made which trade off the risk of infection with the reward of normal activity. This paper addresses the challenge of making such decisions, and provides a practical simplified type of analysis. The question is whether the simplified analysis is fit for practical purpose.

The simplified example selected for detailed study is air travel. As a former safety consultant for a major international airline, I know that all major airlines will have undertaken their own far more elaborate safety studies to make flying as COVID-19 secure as possible. These studies will include tests of aerosol dispersion within planes, and the practical effectiveness of sanitization measures.

Almost 200 passengers were on board a flight from the Greek island of Zante to Cardiff, Wales, on August 25. As many as seven people from three parties were infectious on the plane. Sixteen have since tested positive for the coronavirus. According to passengers, the flight attendants did not enforce the COVID-19 rules sufficiently. The largest concern over flying is being on a plane where a group of younger asymptomatic passengers may flout the rules about keeping their face coverings on, instead leaving them round their chins. Flouting of rules is itself a contagious mode of human behavior. Once a number of passengers remove their face coverings, others will follow. Passengers may also wander around the cabin to talk with friends without face coverings.

Disregard of COVID-19 rules on board a plane is not taken into account in the authors’ air travel example. For any mass gathering of people, e.g. stadiums, disregard of COVID-19 rules is a serious worry. Over the summer, outdoor sports stadiums could have allowed 10% of the usual number of spectators, who could be easily socially distanced. The concern has been over the deliberate violation of social distancing measures.

The authors’ paper should be revised to include a substantial improvement to their air travel example. Recognition of noncompliant human behaviour is essential to be realistic.

Reviewer #3: Review of PONE-D-20-27466 “A deterministic linear infection model…” by McCarthy et al.

The paper seems to argue that there needs to be a way to compute relative risks that is simpler for decision makers to break down and interpret. I agree that this is a worthy avenue of research, as communicating the impact of risk reduction in the aid of proper policy design is crucial to seeing us safely through this pandemic.

However, I believe this manuscript should not be published for the following reasons.

• The paper’s conclusions reflect the input assumptions in a way that makes it appear to be an elaboration of the obvious. It would be more interesting and helpful if the paper’s model were used to compare strategies, demonstrate some surprising conclusion, or compute (believable) quantitative results.

• Simply adding together probabilities, rather than computing the “probabilistic sum”, is called the rare event approximation in risk analysis. This method is used without relevant reference, and in a circumstance where it is arguably inappropriate as the estimated events are not rare. The correct calculation with the probabilistic sum is not much more complicated, and need not necessarily be a barrier in communicating with policy analysts.

• Statements in the discussion such as “Making masks mandatory, and enforcing this rule, is clearly the most cost-effective strategy” are not supported by any of the analysis in the manuscript. I would not dispute this claim, but it is not in any way supported by the example given, nor are most of the other claims in the final paragraph of the discussion where the example is brought up.

• Assumption A6 is a requirement for the proof of A1, and therefore when A6 is being questioned, A1 is being questioned in the discussion. Considering A1 is a foundational assumption for the manuscript this is cause for concern. I agree that this assumption should be questioned, but the paper itself seems to be questioning whether it is valid. The authors begin to acknowledge this issue, but their discussion seems insufficient.

• The independence assumption in A2 and A6 is also perhaps not justified in the example given. For example, the suggested means of boarding back to front without disrupting queuing order would place passengers mostly in close proximity to those they had queued with. Therefore the risk of infection whilst seated would be highly dependent on the risk of infection whilst queuing.

• Assumption A4 notes that the risk of infection depends not only on the distance from the neighbour, but the direction they are facing etc. This assumption is not consistent with the example given, or its application is not apparent. The equation in 3.1 seems to treat all passengers within a queue equally, indicating that the direction they are facing is not a relevant factor.

• Assumption A7 assumes that multiple individuals from the same household do not infect each other and that this assumption is important when applying the model. This assumption is not justified, nor is the implied importance of the assumption ever discussed. Also a point of the discussion notes the impact of households of three travelling together as an additional consideration when leaving middle seats vacant, despite this being perfectly possible whilst maintaining vacant middle seats.

• The model allows for transmission through airborne particulates, touching contaminated surfaces and direct physical contact. Yet, in the example that is given the latter two are assumed to be of negligible impact.

• Use of non-SI units.

• References poorly formatted.

• Non-standard use of pi before definition.

• Equations are not numbered throughout.

• Doesn’t meet the PLOS ONE formatting requirements. For example appendix 8 contains an important definition that would benefit from being in the main body of the text.

• Additional case studies might improve the manuscript. For instance, what would the impact of different boarding policies be? The authors assume that the aircraft will be seated from back to front perfectly, but the impact of this happening imperfectly could be discussed. As could the impact of having an unordered boarding policy.

• The analysis could be improved by having examples where a decision has to be made between two scenarios that are not clear cut. It is fairly trivial to assume that having fewer people on an aircraft would lower the risk of one becoming infected. If there was an example such as having a full school attendance with everyone wearing a mask and limited social distancing versus only having half attendance with full social distancing (>2m) then the model may appear to be more useful.

• The cited literature seems depauperate at best. There seem to be no publications on the subject of disease transmission from grouping on aircraft. They could have cited the recent JAMA article: https://jamanetwork.com/journals/jamanetworkopen/fullarticle/2769383?utm_source=For_The_Media&utm_medium=referral&utm_campaign=ftm_links&utm_term=081820

6. PLOS authors have the option to publish the peer review history of their article (what does this mean?). If published, this will include your full peer review and any attached files.

Reviewer #1: No

Reviewer #2: **Yes: **Gordon Woo

Reviewer #3: No

---

## [Author Response · Author response to Decision Letter 0]

4 Dec 2020

(Word Document version included in uploaded documents).

REVIEWER #1

Summary and Recommendation: 

McCarthy et al. have formulated a “deterministic linear” modeling framework to inform risk- cost-benefit analysis of activities during the SARS-CoV-2 pandemic. The model outputs infection probabilities that are additive over defined sub-activities and infection pathways. Namely, the model assumes independence of the individual probabilities and that activities cannot last longer than one day. Then, the model is applied to estimate the risk associated with taking an airplane ride, under varying parameter choices. Furthermore, the authors argue that the linearity assumption makes the model easier to interpret, with minimal sacrifice in reliability. 

This is a very timely topic and I would like to applaud the authors for tackling this topic using a more abstract, mathematical framework. The results suggest that the model has a high potential utility in estimating relative risks associated with various activities. I would recommend this paper for acceptance, but believe a couple aspects of the manuscript could be thoroughly revised. First, the paper needs further clarity in terms of definitions (i.e., “deterministic” model), assumptions, and presentation of the mathematics. Second, the model application to the airplane ride example should consider a wider range of parameters and augmented interpretation of estimated risk values. These revisions to the paper will enhance the demonstration of the model’s potential ability for application in risk estimation and decision making. 

Reply: Thank you for reviewing our manuscript. As we describe below, we have implemented the changes you have requested and believe the paper has been improved as a result. Thank you for your comments and suggestions.

Broader Comments: 

I. The model is posited as a “deterministic model” throughout the manuscript. It appears that “deterministic” refers to the fact that the values of the model parameters are deterministic (i.e., the 𝜏 parameters in Section 2 and the probability that a neighbor is infected), although this was not explicitly laid out. I believe that more exposition on the meaning of “deterministic” within the context of the model is desirable. In some cases, the usage of the word can be confusing – a few examples below: 

In the Introduction, the authors mention the “advantages of a linear (deterministic) model”, as if “linear” and “deterministic” are directly interchangeable. This is somewhat confusing and should be explained more thoroughly! 

The authors differentiate between a “deterministic or statistical” model in Section 2.0.2. What exactly is the difference in this context? After all, we are still concerned with a probabilistic model, which outputs probabilities. 

Reply: Thank you for your comment. It is clear that our terminology was confusing, and upon reflecting on its use in the passages you referenced and elsewhere, we have decided to change how we refer to the model. We now describe the model as one of “relative risk” – if we pique the reader’s interest and they continue through the paper, they will learn all they need to about the model. Emphasizing that it is not, e.g., a regression model, is probably less essential than we thought it was when we wrote the original draft and used the terminology that we did there. We hope you find the change an improvement. 

II. The authors applied their model to the specific example of COVID-19 risk on an airplane ride. A brief discussion of how estimated risk differs with varying parameters is provided in Section 3.4-3.5. The number of scenarios considered (three varying airplane distancing and boarding distance, three decay assumptions) is relatively small and should be expanded. Ideally, there should be a visualization that demonstrates how varying certain parameters alters the risk (i.e., figures or counter plots). Some more discussion on the interpretation of the discrepancy in estimated risk values (i.e., 359 for scenario 1 vs. 146 for scenario 2) is also desirable. 

Reply: Following the reviewer’s suggestion, we have expanded the airplane example, including its sensitivity analysis section, and added the new Figure 2, which shows how risk changes as a function of flight time, seating arrangement, and decay model. Inspired by the reviewer’s comments, we have also added additional examples, which include an analysis of attending a sporting event and of classroom attendance (at the level of detail of the air travel example in the original manuscript), and then sketches of applying the same approach to restaurants and religious services. We have also included more text about comparing the different numbers produced by the different models.

Specific comments on manuscript details: 

ABSTRACT 

• The authors state early on that their model “can produce estimates of relative risk”. This is an important point – the distinction between relative and absolute risk, as well as the value of estimating relative risk should be more clearly laid out here. 

Reply: Thank you for the suggestion – we have made changes to the abstract along the lines that you suggest, including noting in the abstract that relative risk allows one to rank sub-activities by their risk of infection even when the probability of infection is unknown (or highly uncertain).

• “..in which we assume infections come from a series of independent risks.” Without having read the entire article, it was not immediately evident to me that “series of independent risks” refers to independence with regards to time interval, mode of transmission, etc. I would consider discussing the independence property earlier in the abstract, and provide some more explanation (of course, the full treatment is still most appropriate for Section 2). 

Reply: Following the reviewer’s recommendation, we rephrased the sentence about independence in the abstract. However, given that the abstract must be concise, we leave the more thorough discussion to Section 2.

SECTION 2 

• Section 2.0.2, A7: In my opinion, this assumption requires more discussion (which was included to some extent in Section 4.1). If a household member is infected, this member presumably has a very high likelihood of infecting other household members upon returning home. This is because household members spend significant time together indoor, share surfaces, and are perhaps rather likely to not wear PPE while at home. 

Ideally, the model would be extended to account for the possibly significant increase in risk from household members doing activities together. If this is not feasible, at least some further discussion may be desirable. 

Reply: Thank you for the comment. Once somebody is infected, there is indeed a significant chance that they will infect another household member. But it is unlikely that I would catch COVID from a household member while attending a baseball game together, and not have caught it if we both stayed home. We have added more about household transmission to the discussion.

• Section 2.1: Here, a couple examples relating 𝑠, �, and 𝜋 are provided. The authors give a couple estimates of 𝜋 in the Appendix using real data. I would suggest listing these estimates in Section 2.1, so that the reader has some more context regarding realistic values of 𝜋. 

Reply: We have added some of those numbers from the Appendix to the main text.

• - Section 2.2: The model for infection probability by air-borne transmission includes the term [time of � ], so it appears that infection probability scales linearly with time spent in the vicinity of an infected neighbor. 

I would suggest adding an assumption (or multiple assumptions) in Section 2.0.2 to differentiate between additive and multiplicative properties for the probabilities. This would account for the linear relationship of the probabilities with [time of �j ].

Reply: Our apologies to the Reviewer, but we do not understand the comment. If it has not been addressed among the changes in our revised manuscript, could the Reviewer rephrase it? The probability from each subactivity is added together; the probability for each subactivity is calculated in a straightforward way.

• The contention “that this model is strategically valuable even without knowledge of the parameters...” should at least be briefly justified. 

Reply: We have added some extra text after that claim along the lines suggested by the reviewer.

SECTION 3 

• In the sub-activity decomposition, one sub-activity considered is ‘duration of the flight’. I think that activities such as passengers walking around, using the restroom, and interfacing with flight attendants (such as during meal service), etc. need to be accounted for. Or if such activities are insignificant from the risk point-of-view, the insignificance should be justified. 

Reply: This is totally correct - to fully analyze the flight, all the sub-activities would need to be considered separately. Space limitations preclude us from doing this for each of our examples, but in Example 3.2 we do consider the effect of eating while attending a sporting event. We have also calculated the risk at a stadium from going to the bathroom, but the calculation is lengthy, and the risk relatively small, so we did not include it, but we could if the editor would like us to. We have also included more discussion of the limitations of our examples, including discussing the kinds of details mentioned by the Reviewer, which would be required for the fine-grained analysis that a decision-maker would want to complete for their specific decision context.

• Sections 3.2-3.3: The calculation of the risk contributions (1.14 in Section 2, 1.84 in Section 3) needs more clarity (either here or in the Appendix). It is not immediately evident from the text how these numbers were obtained. 

Reply: We have included more details in Appendix 8.

Furthermore, the risk in Section 3.2 is calculated to be 27.58. But in Section 3, the authors chose to round this number up to 27.6 It is best to be consistent with significant figures. 

Reply: Thank you for catching that discrepancy in significant figures – we have fixed it (and others). In the calculations, we included more decimal places to make the calculation easier to follow. In reporting the final risk score, we rounded to the nearest integer. We have added more detail to show the calculations in the text.

• Section 3.5: The authors claim the second decay function “has a very rapid decay” – best to provide a quantitative justification for “very rapid” here. 

Reply: Thank you for the comment. We have included the quantitative decay from 1 to 2 meters.

SECTION 4 

• The authors claim that doubling the cleaning frequency will approximately halve the probability that the surface is contaminated. This should be mathematically justified (1- 2 sentences is sufficient if the math is trivial). 

Reply: We have included a mathematical justification.

• It was an excellent idea to include discussion of perceived vs. actual risk. The section could benefit from a few more sentences on how to model and quantify perceived risk, as well as discrepancy between perceived and actual risk. Regarding a decisions-making framework, some more quantitative discussion on how perceived risk could be incorporated is also beneficial. 

Reply: Thank you for the suggestion. We have added text about approaches to modeling perceived risk, such as the mental models approach of risk communication (Morgan, Fischhoff, Bostrom, & Atman, 2002) and approaches from cognitive science (Slovic, Finucane, Peters, & MacGregor, 2004), as well as psychophysics (Fox-Glassman & Weber, 2016). We have also added text conjecturing how one might incorporate such work into a specific decision analysis.

• Section 4.1: The discussion on limitations is solid but could benefit from more discussion on how much the limitations quantitatively affect the model accuracy, bias, or estimation confidence. In addition, are these limitations possible to address within the “deterministic linear” framework proposed, or would addressing them require a fundamentally different model? 

Reply: Thank you for the comment. We have expanded our section on limitations and hope some of your concerns are addressed there. In our review of the literature, the biggest weakness is the lack of data – nobody knows how risk decays with distance or direction, what the relative risks between aerosol and droplet transmission are, what the effect of being outdoors is, what the risk of fomites is, etc. Given that lack of data, we think the focus should be on making sure risks and possible effects of mitigating actions are in the correct direction. 

SECTION 6 

• The correlation factor is defined as 𝑓(𝑠). Some more clarity regarding why 𝑓(𝑠) can be considered the correlation factor is desirable. 

Reply: f(s) is the correction factor, not the correlation factor. We explain more clearly in the revision what we mean by this. 

• In Section 6.2, the authors thoroughly discuss some estimates of 𝜋. However, the link from estimates of 𝜋 to how close 𝑓(𝑠) is to 1 is not entirely evident. Is Table 4, which outputs 𝑓(𝑠) for each considered value of 𝜋, important in this regard? This need to be more clearly discussed. 

Reply: We added an explanation of how the tables can be used in Section 6.1.

SECTION 7 

• Why omit the proof? Even if the proof is trivial, one should include a few lines sketching how the result is obtained. 

Reply: The proof has been added.

Our thanks again to the reviewer for their close reading of our manuscript and their comments. 

 

REVIEWER #2

Reviewer #2: During the COVID-19 pandemic, difficult decisions are being made which trade off the risk of infection with the reward of normal activity. This paper addresses the challenge of making such decisions, and provides a practical simplified type of analysis. The question is whether the simplified analysis is fit for practical purpose.

The simplified example selected for detailed study is air travel. As a former safety consultant for a major international airline, I know that all major airlines will have undertaken their own far more elaborate safety studies to make flying as COVID-19 secure as possible. These studies will include tests of aerosol dispersion within planes, and the practical effectiveness of sanitization measures.

Almost 200 passengers were on board a flight from the Greek island of Zante to Cardiff, Wales, on August 25. As many as seven people from three parties were infectious on the plane. Sixteen have since tested positive for the coronavirus. According to passengers, the flight attendants did not enforce the COVID-19 rules sufficiently. The largest concern over flying is being on a plane where a group of younger asymptomatic passengers may flout the rules about keeping their face coverings on, instead leaving them round their chins. Flouting of rules is itself a contagious mode of human behavior. Once a number of passengers remove their face coverings, others will follow. Passengers may also wander around the cabin to talk with friends without face coverings.

Disregard of COVID-19 rules on board a plane is not taken into account in the authors’ air travel example. For any mass gathering of people, e.g. stadiums, disregard of COVID-19 rules is a serious worry. Over the summer, outdoor sports stadiums could have allowed 10% of the usual number of spectators, who could be easily socially distanced. The concern has been over the deliberate violation of social distancing measures.

The authors’ paper should be revised to include a substantial improvement to their air travel example. Recognition of noncompliant human behaviour is essential to be realistic.

Reply: We thank the reviewer for their comments. We have changed the text in how we present the example – and have added additional examples at the request of the other reviews -- in order to make it clear that we do are not providing the example(s) as a fully-worked analysis, one that someone could use off the shelf, but rather, for pedagogical purposes, demonstrating where our model might be useful within the context of an analysis that includes the sorts of details that a subject-specialist would know. Therefore, the air travel example is not meant to be the centerpiece of the manuscript, and we have changed the language in the manuscript to better bound the ambitions of our work and note the need for future work. We believe that the addition of the other examples – as well as a spreadsheet that allows readers to change parameter values and see the calculations for themselves – all help to ensure that the examples are understood as intended.

Following the reviewer’s advice, we have added some additional text about the difficulties of incorporating uncertain human behaviour into analyses. We have suggested, following the reviewer and Reviewer #1’s comments, how might one go about the descriptive research necessary to characterize that behaviour.

 

REVIEWER #3

The paper seems to argue that there needs to be a way to compute relative risks that is simpler for decision makers to break down and interpret. I agree that this is a worthy avenue of research, as communicating the impact of risk reduction in the aid of proper policy design is crucial to seeing us safely through this pandemic.

However, I believe this manuscript should not be published for the following reasons.

• The paper’s conclusions reflect the input assumptions in a way that makes it appear to be an elaboration of the obvious. It would be more interesting and helpful if the paper’s model were used to compare strategies, demonstrate some surprising conclusion, or compute (believable) quantitative results.

Reply: Thank you for the comment. We have expanded the section on examples by including more detail in the air travel example – including additional sensitivity analyses – and by including additional scenarios. We also now include a spreadsheet that allows the interested reader to see more detail of the calculations and perform more sensitivity analyses. In our view, a quantitative model’s usefulness is not diminished when it validates the conclusions of one’s conceptual model. While having surprising results might be more exciting to read, that is a matter of taste. We defer to the Editor about whether to alter our exposition. 

• Simply adding together probabilities, rather than computing the “probabilistic sum”, is called the rare event approximation in risk analysis. This method is used without relevant reference, and in a circumstance where it is arguably inappropriate as the estimated events are not rare. The correct calculation with the probabilistic sum is not much more complicated, and need not necessarily be a barrier in communicating with policy analysts.

Reply: A main point of the paper is that there are great advantages in simplicity and communicability in using an additive model – conceptually, this is much easier to convey to the general public. Moreover, we prove mathematically in Section 6 that the difference between the sum of the probabilities and the exact mathematical formula is small, and far less than the inherent errors in estimating the probabilities in the first place. 

• Statements in the discussion such as “Making masks mandatory, and enforcing this rule, is clearly the most cost-effective strategy” are not supported by any of the analysis in the manuscript. I would not dispute this claim, but it is not in any way supported by the example given, nor are most of the other claims in the final paragraph of the discussion where the example is brought up.

Reply: Thank you for the comment. We have changed the exposition in the paragraph, adding more detail to explain when our example analysis supports a given conclusion, and changing the language when we are speculating. We should have qualified those statements in the original manuscript. Our thanks for pointing out our error. 

• Assumption A6 is a requirement for the proof of A1, and therefore when A6 is being questioned, A1 is being questioned in the discussion. Considering A1 is a foundational assumption for the manuscript this is cause for concern. I agree that this assumption should be questioned, but the paper itself seems to be questioning whether it is valid. The authors begin to acknowledge this issue, but their discussion seems insufficient.

Reply: Thank you for your comment. We have clarified the invocation of A6 in proving that $s$ is a good approximation to $p$. We have added more to the discussion in the limitations, following the reviewer’s suggestion. 

• The independence assumption in A2 and A6 is also perhaps not justified in the example given. For example, the suggested means of boarding back to front without disrupting queuing order would place passengers mostly in close proximity to those they had queued with. Therefore the risk of infection whilst seated would be highly dependent on the risk of infection whilst queuing.

Reply: Thank you for the comment. This is a good observation, but we do not think it violates independence. Of course, there is more than one way to define independence; our interpretation is as follows: We are assuming that the probability of becoming infected during the whole activity is sufficiently low that we can ignore the chance of the event happening to the same person twice. If I am standing and then sitting beside some person X who is infectious, my probability of becoming infected on both occasions is indeed higher than if my immediate neighbor Y were non-infectious. But the probability that I become infected from X while boarding is independent of the probability of becoming infected from X while sitting. It is true that independence could be interpreted more strictly, in which case the fact that I am standing beside X while boarding does affect my probability of sitting beside X. But if X is the only infectious person on the plane, the probability they infect someone is approximately the sum of the probability they infect someone while boarding and the (much larger) probability that they infect someone while sitting. It does not really matter if the two potential targets are the same or different. 

• Assumption A4 notes that the risk of infection depends not only on the distance from the neighbour, but the direction they are facing etc. This assumption is not consistent with the example given, or its application is not apparent. The equation in 3.1 seems to treat all passengers within a queue equally, indicating that the direction they are facing is not a relevant factor.

Reply: This is because there is no good data on how infection varies with direction (as we see, there is no agreement even on how it varies with distance, as the variation between the Chen and Chu model show). Once research has established a better understanding of how infection risk varies with distance and direction, this could be incorporated into modelling boarding. For now, we have to make due with rough estimations.

• Assumption A7 assumes that multiple individuals from the same household do not infect each other and that this assumption is important when applying the model. This assumption is not justified, nor is the implied importance of the assumption ever discussed. Also a point of the discussion notes the impact of households of three travelling together as an additional consideration when leaving middle seats vacant, despite this being perfectly possible whilst maintaining vacant middle seats.

Reply: Thank you for the comment. We have added more discussion about A7 to the manuscript, nothing the importance of household transmission and ways to mitigate it. 

• The model allows for transmission through airborne particulates, touching contaminated surfaces and direct physical contact. Yet, in the example that is given the latter two are assumed to be of negligible impact.

Reply: Airlines claim that they filter the air so efficiently that airborne transmission is not a risk. This may not be correct. In Example 3.2 we consider how to incorporate aerosols. There is no data we know of that gives any ways to numerically compare risks between fomite transmission and air-borne transmission. Until such data becomes available, we don’t see how to combine these into a common score. Direct physical contact with strangers is avoidable outside of hospital settings; again there is no data comparing it to other forms of transmission, but for public events such as the ones in our example this is less important than fomite transmission because it is avoidable. 

• Use of non-SI units.

• References poorly formatted.

• Non-standard use of pi before definition.

• Equations are not numbered throughout.

• Doesn’t meet the PLOS ONE formatting requirements. For example appendix 8 contains an important definition that would benefit from being in the main body of the text.

Reply: Thank you for the comments. We have changed pi to the Greek letter nu – our apologies if the use of pi made the exposition less clear than it could have otherwise been. Our manuscript is typeset using LaTeX, so should we be fortunate enough to be accepted for publication by a journal, we can easily reformat the text to the requirements set out by that journal’s production editor. We do not use SI units because in the US guidelines are given in feet and minutes, so these are natural units to use when quantifying risk. (It is also a little unnatural to think of a flight length or game time in seconds).

• Additional case studies might improve the manuscript. For instance, what would the impact of different boarding policies be? The authors assume that the aircraft will be seated from back to front perfectly, but the impact of this happening imperfectly could be discussed. As could the impact of having an unordered boarding policy.

Reply: Thank you for the comment. In the revision, we have included more variations in the parameters of the air travel example. We have also added new examples, which include an analysis of attending a sporting event and of classroom attendance (at the level of detail of the air travel example in the original manuscript), and then sketches of applying the same approach to restaurants and religious services.

• The analysis could be improved by having examples where a decision has to be made between two scenarios that are not clear cut. It is fairly trivial to assume that having fewer people on an aircraft would lower the risk of one becoming infected. If there was an example such as having a full school attendance with everyone wearing a mask and limited social distancing versus only having half attendance with full social distancing (>2m) then the model may appear to be more useful.

Reply: Thank you for the comment. We hope that the reviewer will approve of the additional examples (which include an analysis of a classroom), as well as the addition of the Excel program allowing the reader to better explore our examples and their limitations. We have also more clearly bounded the ambition of our examples – they are meant to demonstrate who one might apply the model for first-order estimates, and they are not meant to (nor do they) contain the level of detail necessary for a decision-maker in a particular policy domain (e.g., education) to use the example to make a decision. Rather, we hope they convey how one might use the model, and how it might be useful when combined with the subject-matter knowledge that experts in each particular domain would possess.

• The cited literature seems depauperate at best. There seem to be no publications on the subject of disease transmission from grouping on aircraft. They could have cited the recent JAMA article: https://jamanetwork.com/journals/jamanetworkopen/fullarticle/2769383?utm_source=For_The_Media&utm_medium=referral&utm_campaign=ftm_links&utm_term=081820

Reply: Thank you for pointing us to that article. We have updated the cited literature.

---

## [Decision Letter · Decision Letter 1]

23 Dec 2020

PONE-D-20-27466R1

Modeling the relative risk of SARS-CoV-2 infection to inform Risk-Cost-Benefit Analyses of activities during the SARS-CoV-2 pandemic

PLOS ONE

Dear Dr. McCarthy,

Thank you for submitting your manuscript to PLOS ONE. After careful consideration, we feel that it has merit but does not fully meet PLOS ONE’s publication criteria as it currently stands. Therefore, we invite you to submit a revised version of the manuscript that addresses the points raised during the review process.

One reviewer is critical, please address his concerns to the extent possible

We look forward to receiving your revised manuscript.

Kind regards,

Igor Linkov

Academic Editor

PLOS ONE

Reviewers' comments:

Reviewer's Responses to Questions

**Comments to the Author**

1. If the authors have adequately addressed your comments raised in a previous round of review and you feel that this manuscript is now acceptable for publication, you may indicate that here to bypass the “Comments to the Author” section, enter your conflict of interest statement in the “Confidential to Editor” section, and submit your "Accept" recommendation.

Reviewer #1: (No Response)

Reviewer #2: (No Response)

2. Is the manuscript technically sound, and do the data support the conclusions?

Reviewer #1: Yes

Reviewer #2: Yes

3. Has the statistical analysis been performed appropriately and rigorously? 

Reviewer #1: Yes

Reviewer #2: No

4. Have the authors made all data underlying the findings in their manuscript fully available?

Reviewer #1: Yes

Reviewer #2: Yes

5. Is the manuscript presented in an intelligible fashion and written in standard English?

Reviewer #1: Yes

Reviewer #2: Yes

6. Review Comments to the Author

Reviewer #1: (No Response)

Reviewer #2: COVID-19 is a rapidly evolving crisis, and actual data are being gathered on infection sources to help make the type of decisions considered in this paper. Public policy is being driven by accumulating data on infections in restaurants, schools, places of worship etc.. In respect of sports stadiums, a prime concern has been on the travel to stadiums, some of which may involve public transport, and the mingling of fans in congested bars outside the stadiums.

While the analysis that has been undertaken has some merit as an academic exercise, for practical decision-making, this is rather moot. Indeed, with the deployment of effective vaccines in the coming months, the analysis presented will cease to be of much practical relevance by the end of 2021.

I tend to agree with the third reviewer about the publication of this paper.

7. PLOS authors have the option to publish the peer review history of their article (what does this mean?). If published, this will include your full peer review and any attached files.

Reviewer #1: No

Reviewer #2: No

---

## [Author Response · Author response to Decision Letter 1]

24 Dec 2020

(also included as an attachment)

REVIEWER #1

COVID-19 is a rapidly evolving crisis, and actual data are being gathered on infection sources to help make the type of decisions considered in this paper. Public policy is being driven by accumulating data on infections in restaurants, schools, places of worship etc.. In respect of sports stadiums, a prime concern has been on the travel to stadiums, some of which may involve public transport, and the mingling of fans in congested bars outside the stadiums.

While the analysis that has been undertaken has some merit as an academic exercise, for practical decision-making, this is rather moot. Indeed, with the deployment of effective vaccines in the coming months, the analysis presented will cease to be of much practical relevance by the end of 2021.

Reply: Thank you for reading our manuscript and the comments. Given the variation in vaccine distribution both within and between countries – and the international audience of the journal – we believe that jurisdictions exist where our approach could still be helpful for many months to come. There is also the possibility that the current virus will mutate to a point that the vaccine does not work effectively, or a new pandemic could occur with another air-borne virus. As of 12/24/2020, WHO estimates 1.724 million deaths from COVID-19, and in many places there is significant fatigue with broad lockdown measures. A better understanding of which events are more dangerous, and which mitigations strategies are more effective, could significantly help the formulation of public policy. 

It could be interesting to adjust the model to account for the percentage of vaccinated persons in the population of potential activity participants (e.g., among sports fans). As far as we know, no population exists yet that is sufficiently vaccinated to produce herd immunity. 

REVIEWER #2

Review for publication in PLOS One 

‘Modeling the relative risk of SARS-CoV2 infection to inform Risk-Cost-Benefit Analyses of activities during the SARS-CoV-2 pandemic 

(McCarthy, Dewitt, Dumas, and McCarthy) 

Summary and Recommendation: 

McCarthy et al. have revised their model of SARS-CoV2 relative infection risk, which is used for risk-cost-benefit analyses of typical human activities. Since their first submission of this work to PLOS One, the authors have significantly improved the presentation of their work. The first key change is that the authors have re-posited the model, eliminating reference of the model as “deterministic” and minimizing the ensuing confusion. Second, the authors have significantly expanded the application of their model to human activities. Not only have they improved analysis of the airplane ride case study, but they have also considered several additional environments (stadium, classroom, restaurants, and religious services). 

I would recommend publication of this article, and outline some (mostly minor) revisions for the authors to consider. In particular, I would strongly suggest the authors to consider some of my comments regarding modeling and analysis of risk in some of the microenvironments studied (see Section 3). 

Reply: Thank you for the thorough review of our manuscript and your helpful comments in this and the first review.

ABSTRACT 

• A bit more clarity is desired here regarding the distinction between ‘activity’, ‘route of infection’, and ‘constituent components of activities’. 

• I would suggest clarifying what the ‘linearity assumption’ refers to. This is specified later in the paper, but without further explanation here, it is a bit unclear to the reader. 

Reply: We have altered the manuscript to reflect the reviewer’s suggestions.

SECTION 3 

• The authors claim that the risk from going to the bathroom is small and thus omit the bathroom analysis from the paper. In my opinion, this merits a brief justification. I would suspect there is nonnegligible risk due to the close distance between two people who are using neighboring urinals. 

Reply: We added our calculation of the bathroom risk at one stadium. With less mitigation, the risk would be higher; but it is calculable along the same lines as other parts of the event.

• Regarding long-range aerosol transmission (i.e., as mentioned in 3.2), the authors may wish to consider investigating the CU Boulder COVID-19 Aerosol Transmission Estimator tool. 

Reply: We have included a reference to this model. However, it explicitly excludes calculation of risk from droplets, or any exposure within 6 feet. We know of no published results that allow both droplet and aerosol risk to be included in the same quantitative model, so for now we are using the (admittedly crude) approximation that the aerosol risk is proportionate to the reciprocal of the ventilation per person per unit time. Once data appears that allows both droplet and aerosol risk to be estimated simultaneously, the model should be updated to include these results.

• Regarding Section 3.4.1 (Restaurants), I strongly suggest to discuss the risk associated with indoor vs. outdoor dining (a frequently-debated policy issue facing citywide governments). Ideally this should be accompanied by more in-depth discussion on the ratio of aerosol to droplet risk. 

Reply: We added some remarks on this – basically being outdoors reduces the aerosol risk dramatically, but not so much the droplet risk. We completely agree that an in-depth discussion on the ratio of aerosol to droplet risk would be incredibly valuable. However, the data does not exist yet.

• Some further discussion on modeling of trusted cohorts (i.e., in the context of restaurants and religious services) is desirable in the main body of the text. 

Reply: We explained more clearly what we mean by a trusted cohort. If two people are living together with no attempt at self-isolation, then they incur no significant extra risk from each other if they attend an event together.

SECTION 4 

• The discussion on household transmission is an important addition. How would the model devised by the authors be applied to household transmission risk? Or would this not be possible? Household transmission has been a large contributor to the late-fall spike in infections within the U.S., so I believe that more discussion on this matter would be desirable. 

Reply: Unfortunately, we believe our model cannot be adapted as it stands to households. Households are not “activities” in our sense of the word, because living in one lasts much longer than a day, by its nature. We also believe that the decomposition of living in a household into subactivities would be much more difficult than the other examples, because of the variation in what might constitute a “subactivity,” and the variation in the ability to restrict subactivities, in the way that a stadium operator might do in order to lower infection risk. We have added more to this section to indicate these limitations of our approach.

---

## [Editor Report · Decision Letter 2]

30 Dec 2020

Modeling the relative risk of SARS-CoV-2 infection to inform Risk-Cost-Benefit Analyses of activities during the SARS-CoV-2 pandemic

PONE-D-20-27466R2

Dear Dr. McCarthy,

We’re pleased to inform you that your manuscript has been judged scientifically suitable for publication and will be formally accepted for publication once it meets all outstanding technical requirements.

Kind regards,

Igor Linkov

Academic Editor

PLOS ONE
---

## [Editor Report · Acceptance letter]

6 Jan 2021

PONE-D-20-27466R2 

Modeling the relative risk of SARS-CoV-2 infection to inform Risk-Cost-Benefit Analyses of activities during the SARS-CoV-2 pandemic 

Dear Dr. McCarthy:

I'm pleased to inform you that your manuscript has been deemed suitable for publication in PLOS ONE. Congratulations! Your manuscript is now with our production department. 

Kind regards, 

on behalf of

Dr. Igor Linkov 

Academic Editor

PLOS ONE